# Long-range ferrimagnetic order in a two-dimensional supramolecular Kondo lattice

Jan Girovsky[1,†], Jan Nowakowski[1], Md. Ehesan Ali[2,3], Milos Baljozovic[1], Harald R. Rossmann[1], Thomas Nijs[4], Elise A. Aeby[4], Sylwia Nowakowska[4], Dorota Siewert[4], Gitika Srivastava[1,†], Christian Wäckerlin[5,†], Jan Dreiser[6], Silvio Decurtins[7], Shi-Xia Liu[7], Peter M. Oppeneer[3], Thomas A. Jung[1] & Nirmalya Ballav[8]

Realization of long-range magnetic order in surface-supported two-dimensional systems has been challenging, mainly due to the competition between fundamental magnetic interactions as the short-range Kondo effect and spin-stabilizing magnetic exchange interactions. Spin-bearing molecules on conducting substrates represent a rich platform to investigate the interplay of these fundamental magnetic interactions. Here we demonstrate the direct observation of long-range ferrimagnetic order emerging in a two-dimensional supramolecular Kondo lattice. The lattice consists of paramagnetic hexadeca-fluorinated iron phthalocyanine (FeFPc) and manganese phthalocyanine (MnPc) molecules co-assembled into a checkerboard pattern on single-crystalline Au(111) substrates. Remarkably, the remanent magnetic moments are oriented in the out-of-plane direction with significant contribution from orbital moments. First-principles calculations reveal that the FeFPc-MnPc antiferromagnetic nearest-neighbour coupling is mediated by the Ruderman–Kittel–Kasuya–Yosida exchange interaction via the Au substrate electronic states. Our findings suggest the use of molecular frameworks to engineer novel low-dimensional magnetically ordered materials and their application in molecular quantum devices.

[1] Laboratory for Micro- and Nanotechnology, Paul Scherrer Institute, 5232 Villigen PSI, Switzerland. [2] Institute of Nano Science and Technology, Phase-10, Sector-64, Mohali, Punjab-160062, India. [3] Department of Physics and Astronomy, Uppsala University, Box 516, S-751 20 Uppsala, Sweden. [4] Department of Physics, University of Basel, 4056 Basel, Switzerland. [5] Institute of Physics (IPHYS), École Polytechnique Fédérale de Lausanne (EPFL), 1015 Lausanne, Switzerland. [6] Swiss Light Source, Paul Scherrer Institute, 5232 Villigen PSI, Switzerland. [7] Departement für Chemie und Biochemie, Universität Bern, Freiestrasse 3, 3012 Bern, Switzerland. [8] Department of Chemistry, Indian Institute of Science Education and Research (IISER), Pune 411008, India. † Present addresses: Department of Quantum Nanoscience, Kavli Institute of Nanoscience, Delft University of Technology, Lorentzweg 1, 2628 CJ Delft, The Netherlands (J.G.); Nanoscale Materials Science, Empa, Swiss Federal Laboratories for Materials Science and Technology, 8600 Dübendorf, Switzerland (G.S. and C.W.). Correspondence and requests for materials should be addressed to J.G. (email: jan.girovsky@psi.ch) or to P.M.O. (email: peter.oppeneer@physics.uu.se) or to T.A.J. (email: thomas.jung@psi.ch) or to N.B. (email: nballav@iiserpune.ac.in).

O ne of the intriguing challenges in physics and materials science is to realize long-range magnetic order in low-dimensional materials. There are, however, two major obstacles that inhibit its accomplishment. First, it has been rigorously proven by Mermin and Wagner[1] that the isotropic, direct Heisenberg–Dirac magnetic exchange interactions in infinite systems with dimension $d \leq 2$ cannot lead to a magnetically ordered ground state at finite temperatures. Second, for localized spin magnetic moments surrounded by conduction electrons the Kondo screening effect occurs, which implies that conduction electrons flip their spin and thereby wash out the local magnetic moment[2]. To reach a stable magnetization on the nanoscale, as would, for example, be required for miniaturized spintronic and quantum computing devices[3,4], it is thus desired to develop strategies capable to overcome these obstacles. One possible route is to combine local symmetry breaking caused by a strong substrate-adsorbate hybridization on a metallic substrate with strong spin–orbit coupling of the substrate atoms[5,6]. Doing so enables one to overcome the low-dimensions' constraint and create a high magnetic anisotropy, which is required for its stabilizing effect on the temperature fluctuations of the local spin moment. However, such strong hybridization alters the localized, single spin character of the adsorbate, which would be unfavourable for applications. Alternatively, it has been proposed that magnetic dipole–dipole interaction between spin moments could stabilize long-range ordering at finite, although, very low temperatures[7,8].

A further fundamental interaction that could provide an effective coupling between magnetic moments and possibly redeem the Kondo effect is the Ruderman–Kittel–Kasuya–Yosida (RKKY) interaction[9]. This indirect magnetic exchange interaction involves, similar to the Kondo effect, the conduction electrons[9]. These two competing interactions, simultaneously present in multi-spin systems, provide the ground for an intriguing interplay and a rich phase diagram in three dimensions[10]. In two dimensions, the long-range RKKY interaction between two distant magnetic ad-atoms adsorbed on a metallic substrate has been identified from its typical oscillatory behaviour, that is, a change of the coupling from ferromagnetic to antiferromagnetic depending on the distance[11]. Each of the magnetic ad-atoms embedded in the sea of conduction electrons acts as a scattering centre that induces Friedel-like oscillations in the electron density of the surface and bulk states[12], which thereby provide an effective coupling between the spin moments of the two distant ad-atoms.

The Kondo effect, conversely, acts locally around the magnetic moment and leads to a renormalization of the density of states at the Fermi level. This feature can be resolved as a zero-bias signature by scanning tunnelling spectroscopy (STS), in the vicinity of magnetic atoms and molecules on electrically conducting non-magnetic substrates[13–18]. For such on-surface systems, the simultaneous presence of the short-range Kondo effect and the long-range RKKY interaction will cause competing ground states in the sea of conduction electrons and these interactions thus mutually influence each other[11,17,19].

A key question is whether the RKKY interaction could mediate long-range magnetic order in a low-dimensional multi-spin system in the presence of the Kondo effect. The Kondo renormalization leads to a complete quenching of the magnetic moment for a localized $S = 1/2$ spin moment, but for local moments having spin $S > 1/2$ the underscreened Kondo effect[20] occurs, that is, the local spin moment is only reduced, but not washed out. The remaining non-zero spins could possibly order through the long-range RKKY interaction. Recently, an effort to identify the interplay of Kondo and RKKY interactions has been discussed by Tsukahara et al.[21] who observed a broadening

of the zero-bias feature, detected by STS for iron phthalocyanine (FePc) molecules physisorbed on a Au(111) substrate. They suggested that this broadening could originate from magnetic exchange correlations between nearest neighbours due to Kondo and RKKY interactions. However, the microscopic origin of the broadening could not be unambiguously identified and other mechanisms were also proposed[22–24].

It deserves to be mentioned too that the emergence of long-range ferromagnetic order in a two-dimensional (2D) multi-spin array has been recently claimed for charged organic molecules on a flat graphene/Ru(0001) substrate[25]. The ferromagnetic order was attributed, notably not to the RKKY interaction, but to the direct Heisenberg exchange interaction mediated by overlapping frontier orbitals of the molecules, although this would be at variance with the Mermin–Wagner theorem[1]. Another very recent STS investigation of hydrogen atoms chemisorbed on graphene revealed a magnetic interaction between spin-polarized states on individual hydrogen atoms[26]. Although these are promising steps towards the goal of achieving low-dimensional magnetism, so far, long-range magnetic order caused by the RKKY interaction in a truly one-dimensional or 2D system has not been demonstrated.

Here we introduce a 2D material consisting of two different spin-bearing phthalocyanine molecules adsorbed on Au(111) substrates. Employing the element-selective X-ray magnetic circular dichroism (XMCD) technique, we provide the first unambiguous evidence for remanent long-range ferrimagnetic order obtained on this 2D supramolecular layer. Using STS measurements, we show that the low-temperature magnetic order occurs at temperatures well below the Kondo temperature. We further demonstrate, using ab initio electronic structure calculations, that the long-range order is caused by the RKKY coupling, which can be effective here, because the molecular spins have quantum number $S > 1/2$ and are hence underscreened by the Kondo effect, and can therefore order via the RKKY interaction.

## Results

**Design of the experiment**. We have fabricated a supramolecular lattice consisting of two different metallo-phthalocyanines, hexadeca-fluorinated iron phthalocyanine (FeFPc, Fig. 1a) and manganese phthalocyanine (MnPc, Fig. 1b) by co-deposition on an inert, non-magnetic Au(111) substrate. The Au(111) substrate facilitates molecular self-assembly and, in addition, hosts Shockley-type spin-split surface states that are ideal for promoting RKKY coupling between adsorbed spin moments[27]. The presence of the peripheral fluorine atoms in the FeFPc molecules and hydrogen atoms on the MnPc species directs the self-assembly into a checkerboard arrangement as resolved by scanning tunnelling microscopy (STM) (see Fig. 1c) through the C–H···F–C interactions[28,29]. On this heterogeneous molecular monolayer, we applied a combination of molecule-selective probing (that is, XMCD and STS) of the magnetic moments, which allows us to address the element-specific magnetic moments residing on the long-range ordered Fe and Mn ions, as well as the presence of the Kondo effect individually.

**Topographic images of molecules**. The molecules are easily identified from the inset of Fig. 1c, with the small bright species attributed to MnPc molecules, while the FeFPc molecules appear bigger due to the fluorine ligands. Each molecule has four nearest neighbours of the opposite kind, whereas the next-nearest neighbours are of the same type. The periodic zig-zag-like pattern resolved by STM depicts a characteristic fingerprint of atomically clean Au(111) surfaces, the so-called herringbone reconstruction of the top-most layer. In Fig. 1d, we illustrate the fundamental magnetic interactions involved: the spin-bearing molecules act as

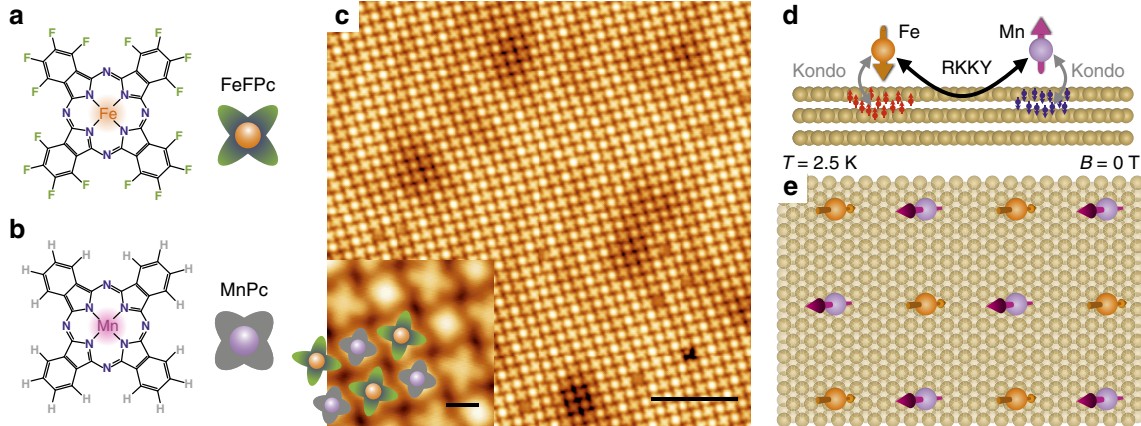

**Figure 1 | Assessing competing fundamental magnetic interactions.** (**a**) Scheme of the FeFPc and (**b**) scheme of the MnPc molecules. The former has all hydrogen atoms on the periphery of the molecule replaced with fluorine. The pictograms shown next to the molecular sketches are used to distinguish the molecules. (**c**) STM image (Bias = − 2.2 V, $I_t = 5$ pA) acquired on an extended domain of FeFPc and MnPc molecules on a Au(111) substrate co-assembled in a checkerboard pattern. Scale bar, 10 nm. The inset shows a zoom of the STM image providing details of the checkerboard pattern with one MnPc species surrounded by four FeFPc molecules and vice versa (scale bar, 1 nm). (**d**) Sketch of the two occurring magnetic interactions in the remanent state, the short-range many-body Kondo screening and the long-range RKKY exchange interaction of the magnetic molecular centres mediated by the conduction electrons of the Au(111) substrate. (**e**) The magnetic moments of the two molecular species are antiferromagnetically coupled and align their moments in the out-of-plane direction.

magnetic centres, to which the spins of substrate electrons couple locally through the Kondo effect, whereas two spin centres can couple via the RKKY interaction mediated by the Au(111) substrate electronic states.

**XAS/XMCD measurements**. We have investigated the magnetic properties of the supramolecular array using element-selective X-ray absorption spectroscopy (XAS) and XMCD, which allows us to probe the magnetic moments of Mn and Fe individually (see Methods). The measurement set-up and our major finding are illustrated in Fig. 2. The applied magnetic field **B**, **k**-vector of incoming X-rays and surface normal are parallel to each other in normal incidence geometry (Fig. 2a). Figure 2b–e shows the XAS/XMCD spectra acquired at normal incidence, at $T = 2.5$ K, and in static magnetic fields of various strengths of 0 and 6.8 T, respectively. In absence of an external magnetic field, that is, for $B = 0$ T (after being magnetized at 6.8 T at 2.5 K), the XMCD spectra demonstrate remanent magnetic moments on both species with out-of-plane orientation (Fig. 2b,c), substantiating the direct observation of long-range magnetic order in a 2D spin-bearing molecular layer. Interestingly, the magnetic moment on the Fe ion of FeFPc molecules is aligned antiparallel to that on the Mn ion of the MnPc molecules, as seen by the opposite signs of the XMCD signals. The antiparallel alignment of Fe and Mn magnetic moments hints at an antiferromagnetic coupling between Mn and Fe ions in the nearest-neighbour positions. Applying a magnetic field of $B = 6.8$ T, both magnetic moments of MnPc and FeFPc are found to align parallel with the field (Fig. 2d,e). The remanent magnetization of FeFPc molecules can also be readily recognized from the XMCD peak height versus $B$ curve as a discontinuity at $B \sim 0$ T (Fig. 2f). The XMCD peak height versus $B$ curve of the MnPc molecules on the other hand, does not display an easily recognizable discontinuity at $B = 0$ T. It is noteworthy that data points in the curve taken at small fields are more susceptible to noise due to the continuous measurement protocol that causes spiky behaviour of the total electron yield (TEY) around zero field (see Methods) and due to excitations induced by the X-ray beam radiation. Although some points close to zero field might suggest a ferromagnetic coupling

(open down triangles in Fig. 2f), the antiferromagnetic coupling and remanence (at $B = 0$ T) of MnPc moments is confirmed by the XMCD spectra shown in Fig. 2c. The antiparallel alignment of the magnetizations on the FeFPc and MnPc sublattices persists for magnetic fields smaller than $\sim 2$ T. At $B = 2$ T, the corresponding Zeeman energy on FeFPc molecules and the long-range magnetic coupling energy becomes comparably large and the net magnetization of the Fe ions goes to zero (Fig. 2f). Applying the mean field approximation to the bipartite Ising model with nearest-neighbour interaction, we estimate the strength of the magnetic coupling from the XMCD peak height versus $B$ curves to be about $J_{\mathrm{Fe-Mn}} = 0.12$ meV, which leads to an ordering temperature of $T_C \sim 3.7$ K. The full lines shown in Fig. 2f depict the fits of the XMCD curves to the mean field approximation model without anisotropy terms; the latter could play a role for small magnetic fields. XMCD spectra measured at $T = 5$ K do not show remanence, which confirms the estimated critical temperature (*cf*. Supplementary Fig. 1f,g). To elucidate the role of the Au(111) substrate on the magnetic ordering, we prepared an FeFPc + MnPc supramolecular array assembled on Ag(111) substrates, which, similar to the Au(111) substrates, host Shockley-like surface states; however, with very different Fermi wave vectors $k_F$ and Fermi density of states[27], crucial for the long-range RKKY coupling, which depends as $J = J_0 \sin(2k_F d)/(2k_F d)^2$ on the distance $d$ between magnetic centres[30]. Our XMCD spectra acquired on the FeFPc + MnPc/Ag(111) system (see Supplementary Note 3 and Supplementary Fig. 4) show no remanent magnetization on either molecules, which suggests the pivotal role of the Au(111) substrate on the emergent long-range order.

The XMCD signals of FeFPc and MnPc measured at normal and grazing incidence (shown in Supplementary Fig. 2) were analyzed with the sum-rule method. In the magnetic field oriented state ($B = 6.8$ T), we resolve sizable induced spin magnetic moments, which are larger when the magnetic field is applied in-plane as compared with out-of-plane (*cf*. Supplementary Table 1). In contrast, the XMCD spectra acquired at remanence in normal incidence geometry are dominated by the orbital magnetic moments ($m_L = -0.39 \mu_B$ for FeFPc and $m_L = 0.28 \mu_B$ for MnPc). The unequal sizes of the moments together with the observed

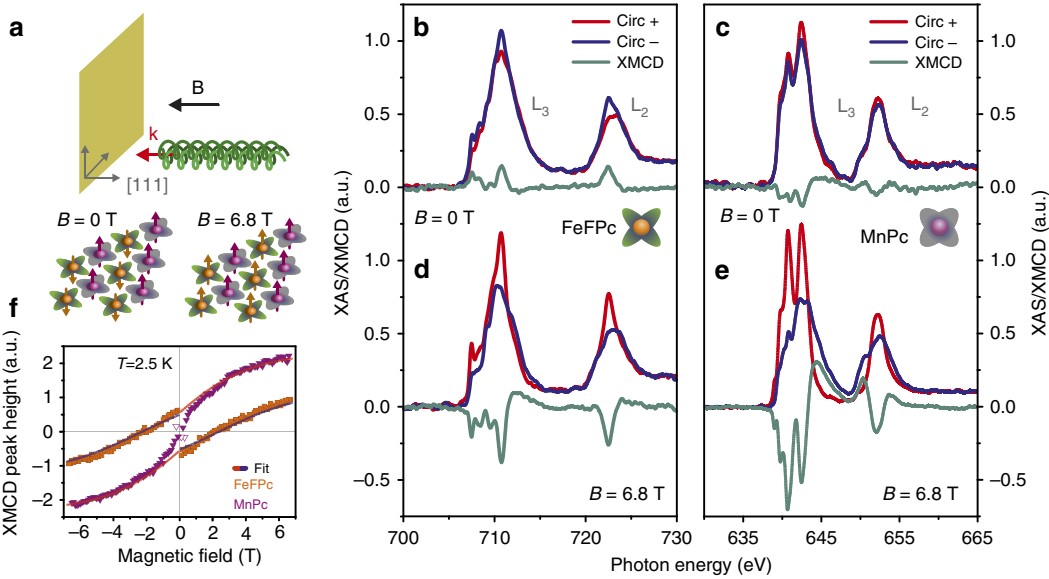

**Figure 2 | Observation of long-range ferrimagnetic order in a 2D supramolecular layer.** (**a**) Sketch of the experimental set-up, in which the XMCD and XAS measurements are performed in normal incidence geometry, with the external magnetic field **B** and the **k**-vector of the X-rays parallel to the surface normal, that is, [111] direction. Illustration of the ferrimagnetically ordered molecular spins in the ground state ($B = 0$ T) and the ferromagnetically aligned spins at $B = 6.8$ T. (**b,c**) XAS/XMCD spectra measured at the Fe $L_{3,2}$ edge and $B = 0$ T demonstrate remanent magnetic moments of FeFPc molecules, which are aligned antiparallel to the remanent magnetic moments observed for the MnPc molecules. (**d,e**) In the applied external magnetic field of $B = 6.8$ T, both molecular spins are aligned parallel to the applied field. (**f**) The measured individual XMCD peak height versus $B$ curves of FeFPc and MnPc molecules show long-range order with antiferromagnetic coupling between the two sublattices. For the FeFPc molecules, the magnetic moment becomes aligned with the applied field for $B \gtrsim 2$ T, as is evidenced by the zero crossing, which appears when the Zeeman energy wins over the FeFPc-MnPc antiferromagnetic exchange coupling. Open down triangles depict the data points at $B \sim 0$ T that possess higher noise level due to the measurement protocol (see Methods). All measurements were performed at $T = 2.5$ K. Full lines show the fit of the XMCD data to a Brillouin function, adopting a mean field approximation without anisotropy terms ($m(\text{Mn}) = 2.3\,\mu_B$, $m(\text{Fe}) = -1.2\,\mu_B$ and $J_{\text{Fe}-\text{Mn}} = 0.12$ meV).

antiparallel coupling establish a ferrimagnetic ground state in the 2D supramolecular lattice. The dominating contribution of the orbital moment can be recognized already from the fact that the $L_3$ and $L_2$ XMCD peaks (Fig. 2b,c and Supplementary Table 1) have the same sign (*cf.* ref. 6). The shape of the XMCD spectra is consistent with earlier XMCD measurements on FePc films[31,32] and on an MnPc thin film[33]. Previously, thick films of FePc molecules were reported to exhibit magnetic anisotropy with in-plane easy axis[31]. Remarkably, here we have observed the first remanent magnetization in the out-of-plane direction for MnPc and FeFPc molecules. Such re-orientation of the predominant magnetization direction is likely to be related to a change of the molecular symmetry upon adsorption onto the Au(111) surface; specifically, here we find a symmetry change from $D_{4h}$ to $C_{4v}$ and $C_{2v}$ for MnPc and FeFPc, respectively (see Supplementary Note 5). A similar change in the magnetic easy axis from in-plane to out-of-plane direction was observed within STS data taken on the FePc species adsorbed on oxygen-reconstructed Cu surfaces[34]. Furthermore, using the XMCD technique a reduction of magnetic moment anisotropy in FePc molecules was observed for the FePc/graphene/Ir system[35]. Our observations suggest that a change in magnetic anisotropy is most likely to be caused by the symmetry reduction; however, other contributions, as for example the hybridization of $d$-orbitals or the balance of in-plane and out-of-plane orbitals of the $d$-electrons should be taken into account.

**STS measurements**. The local Kondo interactions on both molecules have been investigated by the STS technique with the tip positioned above the centre of the molecules. Pronounced Kondo resonances have previously been reported for magnetic molecules on non-magnetic substrates[15,18,21,36], caused by spin flips of the conduction electrons in vicinity of the magnetic

impurity. This many-body renormalization of the energy levels leads to a sharp Kondo resonance, which is observed at the Fermi level for $S = 1/2$ spins[17]. In our case, both molecular species comprise magnetic centres; however, of higher spins. Figure 3 shows differential conductance spectra (d$I$/d$V$) acquired above both FeFPc and/or MnPc molecules around the Fermi energy, that is, around zero bias, in the temperature range of 2.6–9.0 K. The spectra of both molecules exhibit zero bias anomalies, that is, a dip-like feature in the spectra of FeFPc molecules and a step-like shape for the MnPc species, well in line with data reported previously[21,37]. Measurements of differential conductance spectra in the temperature range 2.6–9.0 K result in a smearing and gradual suppression of these features towards higher temperatures confirming their Kondo character[18]. The strength of the Kondo coupling is often expressed in terms of the Kondo temperature that is extracted from a fit of the temperature-dependent spectra. The fit to the STS data using a modified Frota function (for details, see Supplementary Note 4) is shown as red curves in Fig. 3. The fitted line shapes resemble the measured data well. In our case, the Kondo temperature of MnPc and FeFPc molecules is $T_K = 9.4 \pm 1.8$ K and $T_K = 9.2 \pm 2.0$ K, respectively (see Supplementary Fig. 5). These values are somewhat lower/higher compared with those reported for single MnPc ($T_K = 36$ K) and FePc molecules ($T_K = 2.6$ K) on Au(111)[21,37]. Importantly, the main outcome of our STS and XMCD measurements is the emergence of long-range ferrimagnetic order in the supramolecular Kondo lattice at temperatures well below the Kondo temperature of $\sim 10$ K.

**Density functional theory $+\,U$ calculations.** To examine the origin of the observed long-range magnetic order, we have performed electronic structure calculations using the density

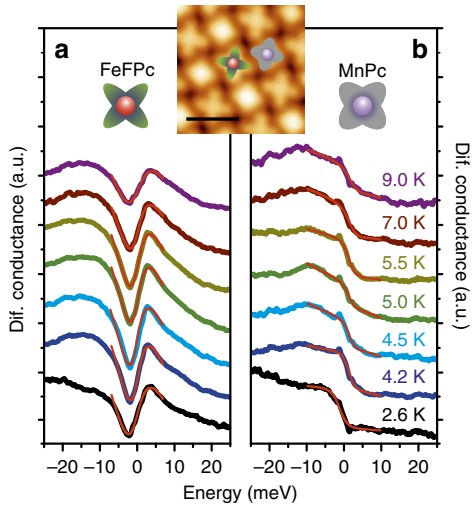

**Figure 3 | STS measurements of FeFPc and MnPc.** (**a**) The temperature-dependent differential conductance d$I$/d$V$ spectra acquired above the centre of the FeFPc molecules show Kondo features around zero bias voltage. The dip-like feature measured on the centre of the FeFPc molecules broadens and becomes shallower with increasing temperature. (**b**) Spectra acquired above the centre of the MnPc species show a step-like shape, which is a signature of the Kondo resonance that broadens and vanishes with increasing temperature. Red full curves are fits to the temperature dependent d$I$/d$V$ spectra with a modified Frota function[17] to determine the Kondo temperatures. The inset shows the area where the spectra where acquired, with a scale bar of 2 nm.

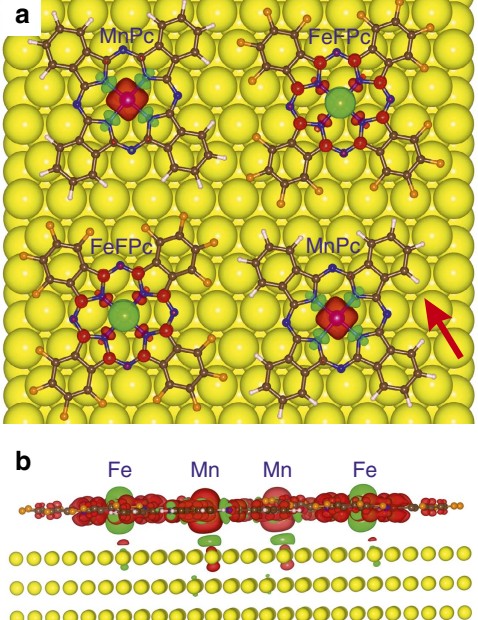

**Figure 4 | Results of density-functional theory-based calculations.**
(**a**) Top-view of the DFT + $U$ computed spin densities of the 2D supramolecular layer on the Au(111) substrate. The red iso-surfaces depict the positive spin density on the Mn atoms of the MnPc molecules and the green iso-surfaces the negative spin density on the Fe atoms of the FeFPc molecules. It is noteworthy that the spin densities on the ligand atoms are opposite to those on the metal centres. (**b**) Side view of the spin density plot at an enlarged iso-density value ($5 \times 10^{-3}$ e$\text{Å}^{-3}$), which shows the interaction of the spin magnetization on the metallo-phthalocyanines with the Au substrate atoms. At this iso-density value, the dominant spin polarization is clearly visible. The direction of the side view is given by the arrow in **a**. Colour code: yellow atoms depict the Au(111) substrate atoms, brown the carbon atoms, blue the nitrogen atoms, white the hydrogen atoms and orange the fluorine atoms.

functional theory with on-site Coulomb $U$ correlations (DFT + $U$) added to capture the strong $d$-electron interactions at the Fe and Mn centres (see Methods). The supramolecular checkerboard pattern is described by a periodic simulation-cell containing $2 \times 2$ molecules, whereas the Au(111) substrate is modelled with three atomic Au layers each consisting of 120 Au atoms. The atomic positions have been optimized by complete self-consistent relaxation of all forces in the 588-atom simulation cell. We find an optimized Fe–Mn distance of 14.35 Å, in agreement with the $13.9 \pm 0.3$ Å, which we measured with STM. Importantly, the calculations predict a ground state with antiparallel coupling between the spin moments on nearest-neighbour FeFPc and MnPc molecules, and a ferromagnetic coupling of each of the species to their next-nearest neighbour. To confirm the lowest total energy of this predicted ferrimagnetic arrangement we have performed total energy calculations assuming an entirely ferromagnetic state and found it to have a higher total energy by 9 K (0.78 meV) for the structural unit cell (that is, one FeFPc + MnPc pair), in accordance with the XMCD measurements. This energy corresponds to a nearest-neighbour Fe–Mn exchange interaction of $J_{\text{Fe}-\text{Mn}} \approx 0.10$ meV (see Supplementary Table 2), consistent with the estimated experimental value. The *ab initio* computed spin density, shown in top view in Fig. 4a, attests the antiparallel coupling between the FeFPc and MnPc spin moments. The computed 3$d$ spin-only moments of $-2.1$ and 3.7 $\mu_B$, for FeFPc and MnPc (see Supplementary Note 5 and Supplementary Fig. 6 for 3$d$-orbital density of state plots), respectively, are in good agreement with those determined by XMCD (of 1.3 and 3.8 $\mu_B$, in the field-aligned state, see Supplementary Note 2 and Supplementary Fig. 3). The measured spin magnetic moment of FeFPc is reduced from the free molecule value $S = 1$ (computed as spin moment $m_S = 2.1\,\mu_B$, DFT + $U$) to $S \sim 1/2$ ($m_S = 1.3\,\mu_B$, obtained from XMCD), which indicates an underscreened Kondo effect in the limit of strong coupling[20]. Our experimental measurements

further provide the practically unperturbed magnetic moment of the MnPc ($S \sim 2$) molecule, $m_S = 3.8\,\mu_B$, which is in good agreement with the value obtained from our DFT + $U$ calculations, $m_S = 3.7\,\mu_B$.

To determine the origin of the ferrimagnetic coupling we have performed additional calculations with the Au substrate removed. Without the Au substrate, no notable magnetic interaction between the neighbouring molecules was obtained, establishing that the molecule–molecule spin coupling is not due to a weak exchange interaction mediated by the overlap of phthalocyanine ligands. Rather, the electrically conductive Au(111) substrate is pivotal for mediating the magnetic interaction between FeFPc and MnPc, which implies that the RKKY interaction is responsible for stabilizing the long-range magnetic order. In Fig. 4b, we show a side view of the calculated magnetization density revealing that the electron density underneath the metal-ion centres exhibits a small spin polarization. This spin-polarized lobe is opposite to the dominant spin polarization resolved on the adjacent molecular centre, evidencing an antiferromagnetic coupling between mobile substrate electrons, predominantly the surface electrons, and the molecular spin.

The magnetic dipole–dipole interaction has previously been proposed as a possible source of coupling between spins that could stabilize long-range magnetic order in pure 2D systems[7,8]. For the here studied supramolecular layer, the dipole–dipole interaction between FeFPc and MnPc moments is however extremely small; for the remanent moments on two nearest

neighbours at a distance of 14 Å, the corresponding dipolar energy is $< 2 \times 10^{-9}$ eV. The dipolar interaction could thus support a long-range ordering temperature of the order of $10^{-5}$ K. This is obviously much less than our observed temperature of a few Kelvin and hence the dipolar interaction can be excluded as source of the observed long-range ferrimagnetic order. Lastly, we mention that a different form of magnetic order in a pure 2D spin system could result from Kosterlitz–Thouless topological order, caused by chiral magnetic vortices[38]. Such topological vortex phase would however have a zero mean magnetization, in contrast to the non-zero XMCD signals that we measured at $T = 2.5$ K.

## Discussion

Previous investigations of the influence of the RKKY interaction on a Kondo system have concentrated on the splitting of the zero-bias resonance[18,20]. Such a splitting has been observed for a Kondo $S = 1/2$ system in an applied magnetic field[18]. A Kondo resonance splitting of an individual MnPc molecule adsorbed on a thin Pb layer on top of a Fe layer has been previously observed and was attributed to an RKKY-type interaction between the molecule and the magnetic iron layer mediated by the conduction electrons in the lead spacer layer[39]. A broadening and splitting of the Kondo resonance has been also observed for FePc molecules on a Au(111) surface, which was tentatively proposed, too, to originate from the RKKY interaction[21]. The unambiguous identification of the involved interactions, however, was compromised by magnetic anisotropies and spin quantum numbers larger than $S = 1/2$ (ref. 40), which complicate the assignment of the Kondo peak splitting to two split $S_z = \pm 1/2$ sublevels. Moreover, the joint appearance of orbital and spin degeneracies leads to other, orbital Kondo effects[22] also affecting the interpretation of the Kondo peak splitting. Furthermore, the broadening of the STS peak of FePc on Au has been attributed to Fe valence fluctuations[23] and, for Fe porphyrins on Au(111), to an adsorption-induced Fe spin switching[24]. Notably, an STS study of how the width of the Kondo resonances of a Co dimer on Cu(001) varies with the Co interatomic distance did not reveal a broadening correlated with their exchange coupling via the RKKY interaction[11]. The latter finding is consistent with our results; specifically, for the here-investigated bimolecular layer we do not observe a clear splitting of the Kondo resonance, even in presence of the observed long-range magnetic order. This can well be related to the fact that the here-studied molecular spins are larger than $S = 1/2$.

Accomplishment of long-range magnetic order in a purely 2D system is an objective that has been actively pursued in recent years[25,26,41]. Garnica et al.[25] deposited a layer of organic molecules (tetracyano-$p$-quinodimethane, TCNQ) on graphene on a Ru(0001) substrate. Employing spin-polarized STS, they observed a spin polarization on TCNQ molecules. As the TCNQ molecules do not react with the graphene, they attributed this to the occurrence of long-range ferromagnetic order due to the direct Heisenberg exchange interaction stemming from intermolecular hybridization of frontier TCNQ molecular orbitals. However, as stated by the Mermin–Wagner theorem[1], this interaction would not lead to long-range ferromagnetic order in two dimensions at finite temperatures. Indeed, an STM image of a somewhat larger section of the TCNQ molecular layer showed magnetic patches with magnetization either up or down, more typical for an overall unordered (paramagnetic) than a ferromagnetically ordered system. Another recent STS/STM investigation of hydrogen atoms chemisorbed on graphene on SiC showed the presence of a spin-polarized state on individual hydrogen atoms[26]. Moreover, a magnetic coupling between the

spin-polarized states on seven different hydrogen atoms was detected[26], which could be a prerequisite to engineer magnetic ordering of hydrogen covered graphene in the future.

The long-range ferrimagnetic order and the many-body Kondo screened state are competing ground states[10,19] for the supramolecular layer. The observed emergence of long-range ferrimagnetic order in the Kondo lattice, at temperatures well below its Kondo temperature, ties in with the spin states of the free FeFPc and MnPc molecules that are larger than $S = 1/2$. Hence, on the Au substrate, the spins of the molecules are underscreened by the conduction electrons, as we measure in particular for FeFPc (which spin is reduced by about $S = -1/2$). The fact that the molecular spins are only underscreened is the key to the appearance of long-range order, caused by the remaining non-zero spin moments that can still order through the relatively weak RKKY interaction.

Our element-specific XMCD measurements provide the first direct evidence for long-range ferrimagnetic order in a binary 2D molecular lattice. This unexpected long-range order of anisotropic moments is observed below the Kondo temperature of $\sim 10$ K and is assigned to the RKKY interaction, mediated by the surface-state electrons of the Au(111) substrate in our experimental and theoretical analyses. Customizing the strength of the fundamental interactions in 2D Kondo lattices by the modification of the molecular building blocks unlocks new possibilities to study low-dimensional quantum phase transitions in synthetic materials and opens up new avenues for building molecular spintronic devices.

## Methods

**Sample preparation.** Clean surfaces of Au(111) crystals have been prepared by Ar ion sputtering ($E = 2$ keV, $t = 20$ min), sputtering and annealing ($T = 500$ °C, $t = 20$ min), and post-annealing ($T = 500$ °C, $t = 20$ min). The cleanness of the surface was checked by X-ray photoelectron spectroscopy, by looking at the most notable contaminants such as carbon, nitrogen and oxygen. The molecules were deposited on the substrates kept at room temperature. The molecules were co-evaporated from tantalum crucibles, which had been pre-heated, and the temperatures of both crucibles were adjusted such that the amount of the molecules measured with a quartz crystal microbalance corresponds to approximately a 1:1 ratio. The 1:1 stoichiometry between the molecules has been verified with the X-ray photoelectron spectroscopy technique by looking at the characteristic fluorine and carbon $1s$ signals typical for fluorinated metallo-phthalocyanines[29].

**XAS/XMCD measurements.** XAS/XMCD measurements were conducted at the X-Treme beamline of the Swiss Light Source at the Paul Scherrer Institute[42]. The XAS and XMCD spectra presented in the main text and in the Supplementary Information were recorded at static magnetic fields. These were conducted at various temperatures using the TEY method of signal acquisition. The size of the X-ray beam spot was $0.5 \times 1$ mm$^2$. The XMCD peak height versus $B$ curves shown in Fig. 2f and Supplementary Fig. 2f were recorded such that the magnetic field was initially set to 6.8 T. At this field we measured XAS/XMCD spectra and decided on a reference photon energy, which corresponds to the most significant peak of the acquired XMCD spectra ($E(MnPc)_{ref} = 641$ eV, $E(FeFPc)_{ref} = 711$ eV). Then, we have chosen the background photon energy, setting it 4 and 5 eV lower than that of the reference energy ($E(MnPc)_{BG} = 637$ eV, $E(FeFPc)_{BG} = 706$ eV), respectively. After that, the field was ramped continuously, while measuring the TEY signal alternatingly at the reference and the background energy. A full loop was repeated twice for the two circular polarizations. The obtained data provide the XMCD peak height value at given magnetic field. In this mode, the reference and background TEY signal are measured at slightly different magnetic fields and the binning of the $B$-field into certain bins gives a difference of $\sim 0.06$ T, with the higher field for the reference energy. It is noteworthy that due to the spiky behaviour of the TEY signal around zero field and the continuous field ramping, with the binning procedure and interpolation, there is increased noise in the XMCD value around zero field.

**STM/STS experiments.** STM/STS experiments were realized with a low-temperature STM microscope (Omicron Nanotechnology GmbH with Nanonis SPM control system) using mechanically cut Pt$_{90}$Ir$_{10}$ tips, which were treated in situ by Ar sputtering and controlled indentation in the bare Au(111) substrate. The bias voltage was applied to the tip and bias voltages given in the manuscript and the Supplementary Information refer to a grounded tip.

STM data were acquired in constant current mode and were processed using Gwyddion software. STS spectra were recorded with open-feedback loop and with initial tip conditions 100 mV/500 pA (lock-in frequency 513 Hz; zero-to-peak amplitude: 1.5 mV).

All the preparation and experimental steps were carried out under ultra-high vacuum (UHV) conditions with the base pressure not exceeding $5 \times 10^{-10}$ mbar.

**Electronic structure theory.** The quantum chemical calculations are based on the DFT $+ U$ approach, in which the strong Coulomb interactions present within the open $3d$-shell of the central metal ion are captured by the supplemented Hubbard $U$ and on-site exchange constant $J^{43}$. In the present calculations, $U$ and $J$ were taken to be 4 and 1 eV, respectively. These values were previously shown to provide the correct spin state for free[44] as well as substrate adsorbed metallo-phthalocyanines and metallo-porphyrins[45,46]. For the DFT exchange-correlation function, the generalized gradient approximation in the parameterization of Perdew et al.[47] was chosen. We used the VASP full-potential plane-wave code[48] with a kinetic energy cutoff of 400 eV. The Au(111) surfaces were modelled by three layers of Au atoms. The Au layers contained 120 atoms each, with two MnPc and two FeFPc molecules on the top layer. The whole simulation cell contained 588 atoms. The total size of the simulation cell was $28.85 \times 29.98 \times 28.71$ Å$^3$. Reciprocal space sampling was performed using $1 \times 1 \times 1$ Monkhorst–Pack $k$-points. We performed full geometric optimizations of the supramolecular FeFPc–MnPc layer on the Au substrate; only the positions of the bottom Au layer were kept fixed and all other positions were free to relax and fully optimized. We find that the four phthalocyanine molecules adsorb on the Au surface with an adsorption energy of 3.5 eV (that is, 0.88 eV energy gain per molecule), which indicates that they are weakly bound to the substrate. The preferential adsorption sites of the metal ion on the top Au layer is in each case computed to be a bridge position and not an on-top adsorption site, consistent with the weak adsorption. The obtained distance between the molecule and the Au substrate is 3.53 Å in the case of MnPc and 3.69 Å for FeFPc, respectively. Further details of the calculations are given in the Supplementary Information.

**Data avalability.** URL link to the depository of all relevant data is available upon request to one of the corresponding authors.

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

## Acknowledgements

We acknowledge the financial support from the Swiss Nanoscience Institute (SNI) (Project P1204), the Swiss National Science Foundation (grants number 20020-162512, 200020-153549 and 206021-121461) and the Swedish Research Council (VR), the Swedish National Infrastructure for Computing (SNIC) and the K. and A. Wallenberg Foundation (grant number 2015.0060). N.B. thanks K.N. Ganesh (IISER Pune) for the support. We thank Rolf Schelldorfer for technical support, Xunshan Liu for assistance with synthesis and Cinthia Piamonteze for experimental support during XMCD measurements.

## Author contributions

J.G., J.N., M.B., C.W., J.D. and N.B. performed the XAS/XMCD experiments. J.G., J.N., M.B., H.R.R., T.N., E.A.A., G.S. and D.S. were involved in the sample preparation and X-ray photoelectron spectroscopy analysis. J.N. and S.N. performed the STS measurements. Molecules were synthesized by S.D. and S.-X.L. XAS/XMCD data have been analyzed by J.G. with help of J.D. STS/STM analysis was performed by J.N. and J.G. M.E.A. and P.M.O. performed the *ab initio* calculations and analysis. J.G. and P.M.O. wrote the manuscript with input from all authors. P.M.O, T.A.J. and N.B. supervised the project.

## Additional information

**Competing interests:** The authors declare no competing financial interests.

**Publisher's note**: 

