## [Peer Review File · Nature Communications]

Reviewers' comments:

Reviewer #1 (Remarks to the Author):

The manuscript contains a series of experiments and theory showing, for the first time according to the authors, long-ranged ferrimagnetic ordering in a 2D system. The combination of two different molecules into such a supramolecular structure is already a remarkable achievement. The experiments and calculations seem to have been properly carried out. The manuscript contains potentially interesting results, but I would expect a deeper analysis of some of the points, as explained below.

I am uncertain about the novelty. The authors claim to be the first ones reporting long-range magnetic order in a 2D system. This claim should be made with some caution since there are similar ones already reported in the past [e.g Nature Physics 9, 368 (2013)] that are at least as credible as the one reported in this manuscript. In fact, the authors discuss the results presented in Nature Physics 9, 368 (2013) in the SI. I believe this discussion pertains to the introduction rather than to the SI if the main claim is to be made. Also, as shown in Nature 531, 489 (2016) or Science 352, 437 (2016) and references therein, long range ferromagnetic order has also been claimed in graphene systems. Still a matter of some debate, but cannot be ignored in the discussion of the novelty of the presented results.

The hysteresis curves presented in Fig. 2D, which are the main result of the paper, need a bit of a deeper discussion. At zero field there is a clear discontinuity in the magnetization curve for FeFPc, but not for MnPc. In fact, maybe looking closely, there seems to be a bit of remanence and coercitivity visible for the latter. Being the remanence of the MnPc molecules so small (if any), how can one explain the negative magnetization of the FeFPc molecules? Why doesn't one observe coercitivity for these molecules and only a discontinuity when the MnPc molecules seem to present a small one?

The evolution of the step-like Kondo feature of the MnPc molecules with temperature can barely be appreciated in the range of experimental temperatures. However, the authors claim a Kondo temperature of 12 K which is near the maximum temperature explored. Showing the actual fittings to the Frota functions might shed some light on this claim. Something similar can be said about the FeFPc molecules which do not seem to show a strong evolution of the zero-bias anomaly with temperature.

The competition between the Kondo screening and the RKKY interaction is not properly discussed. I imagine that these molecules present underscreened Kondo effect where a part of the large magnetic moments of the molecules get Kondo screened, while the remaining not-screened spins are responsible for the RKKY interactions. Otherwise one would not expect to observe both Kondo screening and remanent magnetization at the same time.

The authors attribute the antiferromagnetic nearest-neighbor coupling to the specific nature of the surface states of Au since they do not observe remanence for Ag. They have performed calculations for Au to support this, but this claim should be further supported by calculations for Ag surfaces showing a lack of coupling.

Reviewer #2 (Remarks to the Author):

A Summary of the key results:

The authors present a molecular two-dimensional system in which Kondo physics and long-range magnetic ordering is observed. They demonstrate this using a combination of experimental

methods namely Scanning Tunneling Spectroscopy to identify the Kondo effect and X-Ray Magnetic Circular Dichroism to identify the magnetic ordering. In the chosen system the authors observe a ferrimagnetic order with out-of-plane magnetic moments. The results are interpreted with the help of first-principle DFT calculations which provide insight into the relevant interaction mechanism (RKKY) and the magnetic properties of the adsorbed molecules.

B:Originality and interest:

This specific molecular system (a combination of two phthalocyanine molecules with magnetic moment that assemble in a chessboard pattern) has not been investigated before and neither have similar systems shown to exhibit long-range magnetic ordering in the presence of the Kondo effect. The latter however is known from a variety of (bulk) solid-state systems (heavy fermion systems). In terms of originality one has to concede that it may have been a lucky choice of system that show the presented behaviour the approach to look at a molecular magnetic system is not new. However the interest in the properties of the system will to my mind be substantial as it may serve as a starting point for the development of similar model systems where properties may be tuned.

C:Data & methodology:

The quality of the data and methods of data taking are of highest standards.

D:statistics & uncertainties:

no use of statistics. Uncertainties are given where appropriate and appear plausible.

E: Conclusions

The conclusions appear valid and reliable. The proposed improvements and posed questions in section F should help to guide the readers to maybe more easily access the contents.

F: Suggested improvements:

I have a few detailed comments to make that need no additional data taking or evaluation. The authors should consider and respond to these before publication of the paper.

Intro:32-44:

Although the Mermin-Wagner theorem is undisputable valid one should mention that it is so for infinite systems with isotropic interactions. Which means one should clearly state the reason why the present system can be long-range ordered nevertheless. A similar "discrepancy" to the M-W theorem occurred in the work of Gambardella, et al., Nature 416,301 (2002)

I feel a loose end / severe break in the line of thought from the second to the third sentence of the intro. Maybe it would help first to establish which magnetic interactions may produce long-range ordering and then discuss the Kondo effect.

I. 44 "adsorbate"

I. 56 in "true magnetic 2D systems" or similar - not thin films!

I. 64 Ref 9 is wrong at this point I suggest ref 13 from supplement

Fig. 1 Fig.4 II 72-80

In the figures it does not become obvious what type of superstructure the molecular network forms wrt. the Au(111). Neither in the drawing 1c nor in 4a do the molecules of one type reside on equivalent adsorption sites.

The supercell of the calculation should be given wrt. to the Au(111) and not merely as "containing 2x2 Molecules".

Wouldn't the adsorption site influence the coupling of the central metal atom to the surface electronic states ?

Further quite obviously the Au(111) reconstruction persists underneath the molecules. Does that

influence the Kondo- or RKKY coupling? Probably not, but that needs to be established and written down somewhere.

I. 97: The "discontinuity" comes surprising to me. If the field is ramped from -7 T continuously to 7T the magnetisation should be continuous as well, or? Why should the remanent magnetisation switch? I can only imagine that figure 2d is composed of two measurements ramping the magnetic field from its pos. or neg. maximum to zero field. But that is written nowhere. (The discontinuity is then a measurement artefact, the data should be presented differently.)

I 124: ...beyond symmetry reduction "like" ? the hybridization.. Sentence is not clear to me.

I 129: Citations seem unsuitable, more are in ref list.
Consider also citing Zhang, et al., Nat. Commun. 4, 2110 (2013)

I 132: The fact that $S > 1/2$ should have some consequences
also II. 149-159 is unsatisfactory since the problems mentioned should also concern the present experiment. In that view, what do the authors deduce from their experiments?

I. 163/4: give the supercell wrt. the Au(111) slab

I. 166: "We find an optimized distance..." this makes only sense if the supercell is given. I understand that both molecules are left free to relax? More or at least equally interesting are the optimized adsorption sites of the molecules e.g. the position of the central metal atom wrt. the Au(111) Surface. What about van-der-Waals interactions? Do the calculations produce an energy gain at adsorption? To my mind the bond length of 3.5 or 3.6 AA seems a bit large. I would expect that a true bond to the substrate would require shorter distances that the molecule only adopts due to the vdW interactions.

II. 172/3 Add one sentence to clarify what the (expected) effect of the Kondo screening on the measured magnetic moments should be (measurements are taken $< T_K$) and discuss the values!

II. 183-184: I find "...reminiscent of local Kondo coupling" misleading. The Kondo effect as I understand it, has nothing to do with the static antiferromagnetic coupling between molecule and substrate calculated here. What is true however is that an antiferromagnetic coupling between localized moment and host becomes apparent here that is an ingredient for the Kondo effect. Further: in the supplement the authors argue that the AFM coupling is produced by RKKY via the surface state. I believe this is true. However, in the calculations (with the given slab thickness) the properties of the surface state are certainly not captured well. Which states are responsible for the AFM coupling in the calculations and does that fit to the notion of an oscillatory RKKY coupling with the wavelength corresponding to the intermolecular distances? (as suggested in the supplement)

II.186 conclusions:

given the sentence in the abstract: "...2d systems.. has been elusive, mainly due to the Kondo effect.." and the manuscript title it is necessary to comment here why despite the Kondo-Effect the present system is a 2D magnetic system. To my mind this requires some quantification (or at least indication) of the amount of Kondo screening in the system.

Supplement:

I 61: "overestimate... by a factor of 0.3." Sounds wrong to me a factor of 0.3 makes a value smaller. The authors probably mean "by 30%" but I couldn't establish whether this is the result from Ref 8. At any rate state clearly (maybe in the caption to the table how the values were treated)

Eq. 3 and 4: It is a bit unfortunate to use alpha for an angle that is named theta throughout the manuscript.

G: References:

previous work is cited appropriately, however the authors should carefully check whether the references in the text match the numbering in the list. See improvements in section F

H: Clarity and context:

I see some room for improvement outlined in section F.

Reviewer #3 (Remarks to the Author):

The authors claim to have observed ferrimagnetic order in checkerboard organized Mn and Fe Pc molecules. This claim is interesting and potentially deserves publication in Nature Communications. I, however, see several conflicting observations and open questions in this system that speak against the claim or at least need a more thorough investigation. Thus, I cannot recommend publication of this work in the present form. Both, the presented data and the interpretation are not solid enough. Below, I list the points in more detail.

1. The authors observe with STM a sharp dip at the Fermi energy. This alone is no proof for a Kondo effect. First of all, the authors should plot their fits to the data in the same figure. From visual inspection, I see strong deviations from the measured data and the expected line shape.

2. The authors estimate Kondo temperatures for both molecular species that are 10K and higher. This implies that below 10K, the magnetic moments of the molecules are quenched. Why should one then observe ferrimagnetic ordering at 2.5 K, then? In my view, these are competing ground states and only one of them can possibly be present.

3. The authors have measured the element specific magnetization curves with XMCD. If indeed there was antiferromagnetic coupling between the two species, why does the Fe signal show a sudden jump at 0T but the Mn not? Should they not both flip at 0T at the same time for a ferrimagnetic state? The Mn signal has no jump and looks more like a paramagnet to me.

4. In contrary to the claim by the authors, Figure 2d shows no obvious hysteresis. I suggest that they also show a zoom around 0T that clearly displays a remanence and a coercivity to proof their claim.

5. It seems that a field of about 2T is needed to align both species in the same direction. This gives quantitative information of the coupling strength. From this, also ordering temperatures can be estimated. How does it compare with the DFT calculations? How big is the exchange coupling to nearest and next nearest neighbors there?

6. Are there any experimental observations on the ordering temperatures at all? How does the magnetization curves change with temperature? What happens, when you go above the Kondo temperature?

7. Why does for the Mn XMCD signal at 0T the maximal signal appear at the high energy position of the L3 absorption edge while at 6.8T the maximum is the second highest energy peak. The two curves do not seem to be proportional, which would be a necessary condition for a magnetization normal to the surface.

Reviewer #1 (Remarks to the Author):

The manuscript contains a series of experiments and theory showing, for the first time according to the authors, long-ranged ferrimagnetic ordering in a 2D system. The combination of two different molecules into such a supramolecular structure is already a remarkable achievement. The experiments and calculations seem to have been properly carried out. The manuscript contains potentially interesting results, but I would expect a deeper analysis of some of the points, as explained below.

Authors: We thank the reviewer for his/her careful reading of our manuscript and we are happy to read: *“The combination of two different molecules into such a supramolecular structure is already a remarkable achievement.”*

1. I am uncertain about the novelty. The authors claim to be the first ones reporting long-range magnetic order in a 2D system. This claim should be made with some caution since there are similar ones already reported in the past [e.g. Nature Physics 9, 368 (2013)] that are at least as credible as the one reported in this manuscript. In fact, the authors discuss the results presented in Nature Physics 9, 368 (2013) in the SI. I believe this discussion pertains to the introduction rather than to the SI if the main claim is to be made. Also, as shown in Nature 531, 489 (2016) or Science 352, 437 (2016) and references therein, long range ferromagnetic order has also been claimed in graphene systems. Still a matter of some debate, but cannot be ignored in the discussion of the novelty of the presented results.

Authors: We thank the reviewer for the constructive remarks. We are aware of the work by Garnica et al. [Nat. Phys. 9, 368-374 (2013)] where long-range ferromagnetic order has been claimed in a purely organic 2D layer. We had discussed this paper in the SI. We have now mentioned this work in the new introduction and discuss the findings of this paper in the new “Discussion” section of the main text. As we have already mentioned in the SI, we have certain doubts that long-range ferromagnetic order is proven in the work of Garnica et al. Conversely, our claim of novelty arises from the unambiguous observation of long-range ferrimagnetic order.

In the revised manuscript we have now also cited the two publications mentioned by the reviewer (P. Ruffieux et al. [Nature 531, 489 (2016)] and by H. González-Herrero et al. [Science 352, 437 (2016)]) and discussed the latter one in the Discussion section. Note that these publications have not been highlighted in the previous version of our manuscript as they appeared at the time of the submission of our manuscript. We acknowledge that these articles are a remarkable achievement, however neither of them demonstrates the presence of long-range magnetic order. Specifically, the publication by P. Ruffieux et al. does not provide any experimental demonstration of magnetism. The second publication by H. González-Herrero et al. demonstrates magnetic coupling in an adsorbed structure on graphene containing a maximum of 7 H atoms, i.e. short-range correlations are present, but this cannot be extrapolated as proof for long-range order in a 2D structure.

Action taken: The mentioned publications have been cited and discussed in our new Discussion section.

2. The hysteresis curves presented in Fig. 2D, which are the main result of the paper, need a bit of a deeper discussion. At zero field there is a clear discontinuity in the magnetization curve for FeFPc, but not for MnPc. In fact, maybe looking closely, there seems to be a bit of remanence and coercivity visible for the latter. Being the remanence of the MnPc molecules so small (if any), how can one explain the negative magnetization of the FeFPc molecules? Why doesn't one observe coercivity for these molecules and only a discontinuity when the MnPc molecules seem to present a small one ?

Authors: We thank the reviewer for his/her careful analysis of our data. We agree that remanence in the magnetization curve of MnPc is not clearly resolvable, but our XMCD spectra presented in Figure 2c, top, clearly demonstrate remanent magnetic moments on the MnPc molecules. Table S1 of the Supplemental information provides further evidence about the remanent magnetic moments by presenting the measured values of the effective spin and orbital magnetic moments of the MnPc ($m_{SE} = 0.07 \mu_B$, $m_L = 0.28 \mu_B$) and FeFPc ($m_{SE} = 0.14 \mu_B$, $m_L = -0.39 \mu_B$), confirming an antiparallel alignment of the moment on the FeFPc species with respect to MnPc. We add that magnetization curves for both molecules were measured choosing the most significant peak of the XMCD spectra acquired at $B = 6.8$ T. However the highest peak in the XMCD spectra of MnPc acquired at $B = 0$ T does not correspond to that observed at 6.8 T. We speculate that the change in the peak height is caused by the different origin of magnetic moments. At $B = 0$ T condition, the remanence originates mostly from the orbital magnetic moment. At $B = 6.8$ T, the XMCD signal contains a significant contribution from the spin magnetic moment. Thus the absence of the “jump” in MnPc magnetization curve may be caused by the choice of the reference and by technical limitations at small magnetic fields $B < 0.2$ T.

Action taken: We have clarified the issue in the revised manuscript, and changed the axis label in Fig. 2d and Fig. S2d.

3. The evolution of the step-like Kondo feature of the MnPc molecules with temperature can barely be appreciated in the range of experimental temperatures. However, the authors claim a Kondo temperature of 12 K which is near the maximum temperature explored. Showing the actual fittings to the Frota functions might shed some light on this claim. Something similar can be said about the FeFPc molecules which do not seem to show a strong evolution of the zero-bias anomaly with temperature.

Authors: We thank the reviewer for the suggestion to improve the quality and clarity of our manuscript.

Action taken: We have now incorporated the fits to a Frota function into Figure 3. The fit to the Frota function is described in more detail in the Supplementary Information.

4. The competition between the Kondo screening and the RKKY interaction is not properly discussed. I imagine that these molecules present underscreened Kondo effect where a part of the large

magnetic moments of the molecules get Kondo screened, while the remaining not-screened spins are responsible for the RKKY interactions. Otherwise one would not expect to observe both Kondo screening and remanent magnetization at the same time.

Authors: Yes, the reviewer is right that we are dealing with the underscreened Kondo effect (as was mentioned in the SI).

Action taken: We have now extended the discussion around this point in a new paragraph in the main text (page 7-8).

5. The authors attribute the antiferromagnetic nearest-neighbor coupling to the specific nature of the surface states of Au since they do not observe remanence for Ag. They have performed calculations for Au to support this, but this claim should be further supported by calculations for Ag surfaces showing a lack of coupling.

Authors: Please note that these calculations are at the present *limits* of what is computationally possible. There is no previous *ab initio* calculation that has investigated the indirect magnetic coupling of such large molecules on substrates, which is due to the very large size of the system making it almost prohibitively large for calculations. As we believe that the case for Au, which is at the heart of the manuscript, is convincingly demonstrated, we would prefer not to perform further calculations for Ag surfaces now, but hope to perform such calculations in the future.

Reviewer #2 (Remarks to the Author):

A Summary of the key results:

The authors present a molecular two-dimensional system in which Kondo physics and long-range magnetic ordering is observed. They demonstrate this using a combination of experimental methods

namely Scanning Tunneling Spectroscopy to identify the Kondo effect and X-Ray Magnetic Circular Dichroism to identify the magnetic ordering. In the chosen system the authors observe a ferrimagnetic order with out-of-plane magnetic moments. The results are interpreted with the help of first-principle DFT calculations which provide insight into the relevant interaction mechanism (RKKY) and the magnetic properties of the adsorbed molecules.

B:Originality and interest:

This specific molecular system (a combination of two phthalocyanine molecules with magnetic moment that assemble in a chessboard pattern) has not been investigated before and neither have similar systems shown to exhibit long-range magnetic ordering in the presence of the Kondo effect. The latter however is known from a variety of (bulk) solid-state systems (heavy fermion systems). In terms of originality one has to concede that it may have been a lucky choice of system that show the presented behaviour the approach to look at a molecular magnetic system is not new. However the interest in the properties of the system will to my mind be substantial as it may serve as a starting point for the development of similar model systems where properties may be tuned.

C:Data & methodology:

The quality of the data and methods of data taking are of highest standards.

D:statistics & uncertainties:

no use of statistics. Uncertainties are given where appropriate and appear plausible.

E: Conclusions

The conclusions appear valid and reliable. The proposed improvements and posed questions in section F should help to guide the readers to maybe more easily access the contents.

Authors: We thank the reviewer for his/her positive words on our manuscript and appreciate the diligent evaluation of our manuscript and the suggestions provided. We have addressed the reviewer's concerns/comments as mentioned in the following paragraphs.

F: Suggested improvements:

I have a few detailed comments to make that need no additional data taking or evaluation. The authors should consider and respond to these before publication of the paper.

Intro:32-44:

Although the Mermin-Wagner theorem is undisputable valid one should mention that it is so for

infinite systems with isotropic interactions. Which means one should clearly state the reason why the present system can be long-range ordered nevertheless. A similar "discrepancy" to the M-W theorem occurred in the work of Gambardella, et al., Nature 416,301 (2002). I feel a loose end / severe break in the line of thought from the second to the third sentence of the intro. Maybe it would help first to establish which magnetic interactions may produce long-range ordering and then discuss the Kondo effect.

Action taken: In the revised manuscript, we have expanded the discussion on the Mermin-Wagner theorem in the introductory paragraph and have followed the suggestion to mention first the relevant magnetic interactions. We have also improved the readability of the introduction.

I. 44 "adsorbate"

Action taken: Correction has been made.

I. 56 in "true magnetic 2D systems" or similar - not thin films!

Action taken: The sentence has been modified.

I. 64 Ref 9 is wrong at this point I suggest ref 13 from supplement

Action taken: The reference has been corrected.

Fig. 1 Fig.4 II 72-80

In the figures it does not become obvious what type of superstructure the molecular network forms wrt. the Au(111). Neither in the drawing 1c nor in 4a do the molecules of one type reside on equivalent adsorption sites.

Authors: From our DFT+U calculations we find that both species are weakly adsorbed on the surface (physisorbed), without a strong hybridization of molecular orbitals with surface electronic states. This suggests that there is no strongly preferred adsorption site. We note however, that although there are small differences in the positions we find that the central metal ions are in bridge positions and not in "on top" positions.

Action taken: This information has been added to the Methods section.

The supercell of the calculation should be given wrt. to the Au(111) and not merely as "containing 2x2 Molecules".

Action taken: We have added the requested information in the main text and Methods section (each Au layer contained 120 atoms).

Wouldn't the adsorption site influence the coupling of the central metal atom to the surface electronic states ?

Authors: In principle, yes. Different adsorption sites could have a different coupling to the surface electronic states. But we investigate the most stable configuration, both in the experiment and in the DFT+U theory, which is characterized by one particular adsorption site, i.e., in the DFT+U calculations at zero K the adsorption site is not a free, adjustable parameter.

Action taken: We have clarified this in the paragraph on DFT+U calculations on pages 6-7 of main text and in the SI on pages 8-9 .

I. 97: The "discontinuity" comes surprising to me. If the field is ramped from -7 T continuously to 7T the magnetisation should be continuous as well, or? Why should the remanent magnetisation switch? I can only imagine that figure 2d is composed of two measurements ramping the magnetic field from its pos. or neg. maximum to zero field. But that is written nowhere. (The discontinuity is then a measurement artefact, the data should be presented differently.)

Authors: Each magnetization curve presented in Figure 2d is composed of two magnetization curves. The first magnetization curve is recorded continuously every ~ 0.1 T from 6.8 T to -6.8 T. The second one, similarly, is recorded in the fields starting at -6.8 T and ending at 6.8 T. The discontinuity observed in the magnetization curve of the FePc molecules is due to the collective switching of the Fe spins. The spin of MnPc ($S \sim 3/2$) is larger than that of FePc ($S = 1$) and thus the MnPc spins align along the magnetic field first due to the larger Zeeman energy.

At magnetic fields larger than 2 T , the Zeeman energy is stronger for MnPc then for FePc due to higher spin of MnPc. Below that point the antiferromagnetic exchange coupling between MnPc and FePc molecules becomes dominant aligning Fe spins antiparallel to that of MnPc. As we cross $B = 0$ T the moments of MnPc follow magnetic field and FePc thus "flip" their spins.

The moments of the FePc molecules are aligned antiparallel to those of MnPc when the external magnetic field exceeds the coercive field. We estimate the coercive field to be lower than 0.1T (see SI).

I 124: ...beyond symmetry reduction "like" ? the hybridization.. Sentence is not clear to me.

Authors: The sentence has been reformulated: *"Our observations suggest thus that a change in magnetic anisotropy is most likely caused by the symmetry reduction, however, other contributions,*

as for example the hybridisation of d-orbitals or the balance of in-plane and out-of-plane orbitals of the d-electrons should be taken into account."

I 129: Citations seem unsuitable, more are in ref list.

Consider also citing Zhang, et al., Nat. Commun. 4, 2110 (2013)

Authors: We replaced the unsuitable citation by the one suggested by the reviewer.

I 132: The fact that $S > 1/2$ should have some consequences also

Authors: Yes, we agree with this statement.

Action taken: We have modified the manuscript and have added a discussion of the underscreened Kondo effect, which appears here, since the MnPc and FeFPc molecules in the gas phase are $S = 3/2$ and $S = 1$ systems, respectively. The Kondo effect for $S > 1/2$ magnetic impurities leads to an incompletely screened magnetic moment; such moments can still couple through the RKKY interaction which can thus induce long-range order at low temperatures.

II. 149-159 is unsatisfactory since the problems mentioned should also concern the present experiment. In that view, what do the authors deduce from their experiments?

Authors: In the paragraph mentioned by the reviewer we discuss the possibilities that could lead to a splitting of Kondo resonance as reported in the publication by Tsukahara et al (ref. 18). Its origin is not clear; Tsukahara et al. attributed it to RKKY interaction between neighbouring molecules, however no experimental proof was given. The follow up publications tried to explain the observed splitting differently, assuming an orbital Kondo effect or other mechanisms. Conversely, in our experiments we provide a direct proof of interaction mediated by the RKKY coupling, which in addition leads to a long-range ordering observed in our XMCD experiments. In our STS measurements for a supramolecular layer we do however not observe a clear splitting, which might indeed be related to the $S > 1/2$.

I. 163/4: give the supercell wrt. the Au(111) slab

Action taken: We mention now in the manuscript the real-space dimensions of the supercell and also that we used 120 Au atoms per Au layer.

I. 166: "We find an optimized distance..." this makes only sense if the supercell is given. I understand that both molecules are left free to relax? More or at least equally interesting are the optimized adsorption sites of the molecules e.g. the position of the central metal atom wrt. the Au(111)

Surface. What about van-der-Waals interactions? Do the calculations produce an energy gain at adsorption? To my mind the bond length of 3.5 or 3.6 Å seems a bit large. I would expect that a true bond to the substrate would require shorter distances than the molecule only adopts due to the vdW interactions.

Authors: Yes, the molecules are completely free to relax to the optimal positions.

Action taken: We have added the dimensions of the simulation cell and also mentioned that the central metal atoms of the phthalocyanines are in a bridge position and not in an “on top” position. We have added a discussion of van der Waals interactions in the Supplementary Information. We have also added in the Methods section the energy gain obtained in the calculations (0.88 eV per molecule, suggesting a weak adsorption). For a weak adsorption (physisorption) the bond lengths are actually not unusual.

II. 172/3 Add one sentence to clarify what the (expected) effect of the Kondo screening on the measured magnetic moments should be (measurements are taken $< T_K$) and discuss the values!

Action taken: We have added in the revised manuscript: *“The measured spin magnetic moment of FePc is reduced from the free molecule value $S=1$ (computed as spin moment $m_S = 2.1 \mu_B$, DFT+U) to $S \sim 1/2$ ($m_S = 1.3 \mu_B$, obtained from XMCD), which indicates an underscreened Kondo effect in the limit of strong coupling [Nozieres, Ref. 18]. Our experimental measurements further provide the practically unperturbed magnetic moment of the MnPc ($S \sim 2$) molecule, $m_S = 3.8 \mu_B$, which is in good agreement with the value obtained from our DFT+U calculations, $m_S = 3.7 \mu_B$.”*

II. 183-184: I find “..reminiscent of local Kondo coupling” misleading. The Kondo effect as I understand it, has nothing to do with the static antiferromagnetic coupling between molecule and substrate calculated here. What is true however is that an antiferromagnetic coupling between localized moment and host becomes apparent here that is an ingredient for the Kondo effect. Further: in the supplement the authors argue that the AFM coupling is produced by RKKY via the surface state. I believe this is true. However, in the calculations (with the given slab thickness) the properties of the surface state are certainly not captured well. Which states are responsible for the AFM coupling in the calculations and does that fit to the notion of an oscillatory RKKY coupling with the wavelength corresponding to the intermolecular distances? (as suggested in the supplement)

Action taken: We have removed the criticized words. The reviewer is right that it would be of interest to see from the calculations if there is an oscillatory behaviour of the coupling corresponding to the intermolecular distance. However, in view of the huge size of the supercell that would be needed for such an investigation it will take a few years before such calculations can be carried out. We can currently thus only definitely state from our experiments that the antiferromagnetic coupling is not observed for the FePc+MnPc array on Ag substrates.

II.186 conclusions:

given the sentence in the abstract: "...2d systems.. has been elusive, mainly due to the Kondo effect.." and the manuscript title it is necessary to comment here why despite the Kondo-Effect the present system is a 2D magnetic system. To my mind this requires some quantification (or at least indication) of the amount of Kondo screening in the system.

Authors: The key is here the incomplete Kondo screening of the magnetic moments which are larger than $S > 1/2$. Hence, in the case of the underscreened Kondo effect there are magnetic moments present that can order magnetically via the RKKY interaction. The latter interaction is not excluding long-range order, in contrast to direct Heisenberg exchange.

Action taken: We have reformulated our statement clearer in the new "Discussion" section.

Supplement:

I 61: "overestimate... by a factor of 0.3." Sounds wrong to me a factor of 0.3 makes a value smaller. The authors probably mean "by 30%" but I couldn't establish whether this is the result from Ref 8. At any rate state clearly (maybe in the caption to the table how the values were treated)

Action taken: In the revised version of the Supplementary Information we have changed this and now write: "*...by 33% [7,8]. The values of the Mn magnetic moments given in Table S1 have therefore been multiplied by a factor 0.7.*"

Eq. 3 and 4: It is a bit unfortunate to use alpha for an angle that is named theta throughout the manuscript.

Action taken: We have replaced the symbol alpha with theta in equations 3 and 4, keeping now the nomenclature consistent through the manuscript.

G: References:

previous work is cited appropriately, however the authors should carefully check whether the references in the text match the numbering in the list. See improvements in section F

H: Clarity and context:

I see some room for improvement outlined in section F.

Authors: We have included the comments outlined in section F and hope to have improved thereby the manuscript.

Reviewer #3 (Remarks to the Author):

The authors claim to have observed ferrimagnetic order in checkerboard organized Mn and Fe Pc molecules. This claim is interesting and potentially deserves publication in Nature Communications. I, however, see several conflicting observations and open questions in this system that speak against the claim or at least need a more thorough investigation. Thus, I cannot recommend publication of this work in the present form. Both, the presented data and the interpretation are not solid enough. Below, I list the points in more detail.

Authors: We thank the reviewer for his/her detailed reading of our manuscript. In the following paragraphs we respond to concerns and comments raised by the reviewer.

1. The authors observe with STM a sharp dip at the Fermi energy. This alone is no proof for a Kondo effect. First of all, the authors should plot their fits to the data in the same figure. From visual inspection, I see strong deviations from the measured data and the expected line shape.

Authors: We agree with the reviewer that a sharp dip at zero bias is not necessarily a signature of the Kondo effect.

Action taken: In the revised manuscript we have plotted the fits to the STS spectra in the same Figure 3.

2. The authors estimate Kondo temperatures for both molecular species that are 10K and higher. This implies that below 10K, the magnetic moments of the molecules are quenched. Why should one then observe ferrimagnetic ordering at 2.5 K, then? In my view, these are competing ground states and only one of them can possibly be present.

Authors: As mentioned also in the reply to the second reviewer, the key to the long-range order is the fact that we are dealing with the underscreened Kondo effect, i.e. there are magnetic moments present on the MnPc and FeFPc molecules which can order through the RKKY interaction.

Action taken: This explanation has been added to the new "Discussion" section.

3. The authors have measured the element specific magnetization curves with XMCD. If indeed there was antiferromagnetic coupling between the two species, why does the Fe signal show a sudden jump at 0T but the Mn not? Should they not both flip at 0T at the same time for a ferrimagnetic state? The Mn signal has no jump and looks more like a paramagnet to me.

Authors: We agree with the reviewer on this issue. We attribute the lack of a clear jump to the choice of the reference with the magnetization curve measured at the highest XMCD peak in the spectrum acquired at $B = 6.8$ T. The XMCD peak at the same energy measured at $B = 0$ T is however not the highest one, which we believe is the reason of the missing discontinuity. We emphasize however, that we have a clear indication of remanence on the MnPc molecules from the XMCD spectrum taken at $B = 0$ T.

4. In contrary to the claim by the authors, Figure 2d shows no obvious hysteresis. I suggest that they also show a zoom around 0T that clearly displays a remanence and a coercivity to proof their claim.

Authors: Figure 2d of our manuscript does not show hysteresis due to the fact that coercivity of the system is < 0.1 T, which is below experimental resolution of our setup. In order to resolve coercivity in the system we measured XAS/XMCD spectra at $B = 0.1$ T (-0.1 T) with the magnetic field swept up (down) from $B = -6.8$ T ($+6.8$ T), i.e. crossing $B = 0$ T.

Action taken: We show these spectra in the revised Supplemental information.

5. It seems that a field of about 2T is needed to align both species in the same direction. This gives quantitative information of the coupling strength. From this, also ordering temperatures can be estimated. How does it compare with the DFT calculations? How big is the exchange coupling to nearest and next nearest neighbors there?

Authors: We thank the reviewer for this helpful remark. We applied the mean field approximation to bipartite Ising model assuming the nearest neighbour interaction only, in order to extract the strength of the indirect exchange coupling from the magnetization curves. We find the exchange coupling to be about 0.12 meV leading to an ordering temperature of $T_c \sim 3.7$ K. The estimated ordering temperature is in line with our observation where we resolve no XMCD signal for measurements performed at $T=5$ K (now included in SI).

Action taken: This has been added to the manuscript. As for the DFT calculations, we do treat the full experimental system with the 2x2 arrangement of the molecules, and find self-consistently the ferri-magnetic coupling. However, we cannot compute currently the individual exchange coupling of a molecule to next nearest neighbours, as for this we would need larger supercells. The calculation of the nearest neighbour exchange interaction is also nontrivial, as we have to enforce a ferromagnetic order to obtain the size of the interaction.

6. Are there any experimental observations on the ordering temperatures at all? How does the magnetization curves change with temperature? What happens, when you go above the Kondo temperature?

Authors: Yes, there are. We have performed XMCD measurements at 5 K which show no remanence at $B = 0$ T, consistent with the above estimated ordering temperature.

Action taken: We have added a figure with spectra measured at $T = 5$ K in the Supplemental Information. At higher temperatures, above 10 K (comparable to the Kondo temperature), the energy of thermal excitations would be ~ 1 meV, which is already higher than that of the applied magnetic field ~ 0.5 meV. The thermal spin excitations would thus overcome the effect of the magnetic field and wipe out any magnetic interaction. As mentioned earlier, we do not observe the remanence at $T = 5$ K and thus we expect no remanence above the Kondo temperature, $T > 12$ K.

7. Why does for the Mn XMCD signal at 0T the maximal signal appear at the high energy position of the L3 absorption edge while at 6.8T the maximum is the second highest energy peak. The two curves do not seem to be proportional, which would be a necessary condition for a magnetization normal to the surface.

Authors: We thank the Referee for raising this question. We propose that the slightly different shapes of the XMCD spectra could be due to a difference of the spin and orbital polarizations of the Mn centers between the remanent state and the state at high magnetic field. The remanent magnetic moments of both molecules have considerable contributions from orbital magnetic moments, whilst at $B = 6.8$ T the XMCD signal originates mainly from spin magnetic moments residing on different orbitals.

Reviewers' comments:

Reviewer #1 (Remarks to the Author):

I believe the authors have made a very good job answering my questions and, it seems to me, those of the other referees as well. The paper has been considerably improved and I recommend publication in its present form.

Reviewer #2 (Remarks to the Author):

I have carefully read the rebuttal of the authors and the revised manuscript. The authors have carefully considered my remarks during the first review and have taken appropriate action. To my mind the manuscript has improved considerably to a degree where I wholeheartedly recommend it for publication in Nature Comm. as is. The paper will be a very useful contribution to the field of low-dimensional magnetic systems and the identification of relevant interactions between the spin carrying entities.

Reviewer #3 (Remarks to the Author):

The manuscript has been significantly improved. Many of the points of the referees have been addressed adequately, but there still remain some open questions and room for improvement before I would support publication.

1. Discussion about Mermin Wagner theorem (MWT)

It is essential for the claims to correctly handle the MWT. It claims that the ordering temperature vanishes in 2D (and lower) dimensions in an isotropic Heisenberg system, in which the interactions are sufficiently short ranged.

It only needs dipolar interaction or a slight anisotropy by any spin orbit interaction to lift MWT. Both interactions are active here (as also in the study of Garcia et al.). The authors should thus be more moderate in their claims and criticism to prior work.

2. Long range order

The authors claim to have observed not only a remanence but also long range order. The latter is strictly speaking not shown by the data. The data does not exclude domain formation at all. In this respect, also the criticism to the work of Garcia et al. needs to be softened.

3. Size of the exchange

The authors very nicely deduced the size of the exchange interaction from the XMCD loops. It would be nice to plot the expected behaviour as a line into the XMCD data of figure 2d to illustrate the degree of agreement. Further, it seems imperative to compare the experimentally deduced exchange constants to that of the ab-initio calculations. Actually, no quantitative data is given in the text on the latter.

4. Strange XMCD data

I still do not understand the discrepancy in the presented data regarding the sub-lattice magnetizations. The XMCD spectra at nominally zero field in Figure 2b and d speak for an antiferromagnetic interaction at $B=0$. The data plotted in Fig. 2d shows ferromagnetic alignment at

B near zero. This goes together with an uncertainty of 0.1T in the field calibration. Can the authors exclude a large susceptibility and the observed remanence is just due to the bad field precision?

5. Kondo temperature

I like the temperature dependent STS measurements to deduce the Kondo temperature by extrapolation. However, the STS data was obtained using a finite modulation which contributes to the spectral width of the features. It seems that this has not been taken into account. It would be great to do this in a proper way.

6. Ordering temperature

At 5 K, the remanence vanishes (Fig. S2), which is a nice confirmation of the claims. However, the manuscript does not specify, whether the XAS data were also taken after saturating the samples at high fields, as for the 2.5K data.

Reviewer #1 (Remarks to the Author):

I believe the authors have made a very good job answering my questions and, it seems to me, those of the other referees as well. The paper has been considerably improved and I recommend publication in its present form.

Authors: We are pleased by the reviewer's positive words on our work and for recommending publication.

Reviewer #2 (Remarks to the Author):

I have carefully read the rebuttal of the authors and the revised manuscript. The authors have carefully considered my remarks during the first review and have taken appropriate action. To my mind the manuscript has improved considerably to a degree where I wholeheartedly recommend it for publication in Nature Comm. as is. The paper will be a very useful contribution to the field of low-dimensional magnetic systems and the identification of relevant interactions between the spin carrying entities.

Authors: We are happy to hear the positive words from the reviewer about the revised manuscript.

Reviewer #3 (Remarks to the Author):

The manuscript has been significantly improved. Many of the points of the referees have been addressed adequately, but there still remain some open questions and room for improvement before I would support publication.

Authors: We are glad to read that the manuscript has been substantially improved.

1. Discussion about Mermin Wagner theorem (MWT)

It is essential for the claims to correctly handle the MWT. It claims that the ordering temperature vanishes in 2D (and lower) dimensions in an isotropic Heisenberg system, in which the interactions are sufficiently short ranged.

It only needs dipolar interaction or a slight anisotropy by any spin orbit interaction to lift MWT. Both interactions are active here (as also in the study of Garcia et al.).

The authors should thus be more moderate in their claims and criticism to prior work.

Authors: We agree with the reviewer that strong magnetic anisotropy can overcome the Mermin-Wagner theorem for low-dimensional systems at finite temperature. This has been mentioned already in the previous report of Reviewer #2 (specifically, item F) and we have addressed this already in the revised introduction, where we wrote: *“One possible route is to combine local symmetry breaking caused by a strong substrate-adsorbate hybridization on a metallic substrate with strong spin-orbit coupling of the substrate atoms [5,6]. Doing so, enables one to overcome the*

low-dimensions' constraint and create a high magnetic anisotropy which is required for its stabilising effect on the temperature fluctuations of the local spin moment."

The reviewer #3 mentions the possibility of emergence of long-range order through the magnetic dipole-dipole interaction. This interaction was not discussed in our revised manuscript, but we do agree that it deserves to be considered in all relevant magnetic systems as this interaction could, in principle, lead to long-range magnetic order. We have thus added a discussion on the magnetic dipole-dipole interaction and the possible influence it could have on the long-range order observed in the manuscript. We would like to stress, however, that in our system the nearest-neighbour dipolar interaction is of the order of neV (nano eV) considering the magnetic moments of MnPc and FeFPc molecules observed in the remanence. Moreover, there are cancelling contributions due to alternating spin directions that will affect the interaction. Overall, we estimate an ordering temperature due to the dipolar interaction of a few tens of micro Kelvin.

Therefore we are confident that the dipole-dipole interaction is not relevant for the long-range order in our 2D spin array. In the newly revised manuscript we mention and discuss the dipole-dipole interaction as a possibility for long-range ordering in low-dimensional systems and show explicitly that it is not relevant here.

We are not certain to which publication the reviewer is referring when mentioning Garcia et al. as we do not cite any work by Garcia et al. Assuming that the reviewer is referring to the work by Garnica et al. , we wish to point out that the Mermin-Wagner theorem, magnetic anisotropy or the magnetic dipole-dipole interaction are not mentioned in this publication. Instead, this work claims that long-range ferromagnetic order in a 2D organic layer is caused by the direct Heisenberg exchange interaction. Therefore, while we think that the work by Garnica et al. deserves credit as it considered long-range order in 2D, we also believe that we have correctly addressed the open questions related to their work.

2. Long range order

The authors claim to have observed not only a remanence but also long range order. The latter is strictly speaking not shown by the data. The data does not exclude domain formation at all. In this respect, also the criticism to the work of Garcia et al. needs to be softened.

Authors: We would like to mention that the XMCD technique is considered to be capable to observe long-range order due to breaking time-reversal symmetry leading to a remanence with (here) ferrimagnetic ground state. In the absence of long-range correlations, any short-range interaction would cause formation of areas showing a certain spin-polarization, however, on average the total magnetization would be zero. This is clearly not what we have detected.

We would like to add that in the work of Garnica et al., the long-range order has been deduced from local spin-polarized STM and STS measurements on two domains with opposite magnetization. In their case those domains would indeed cancel each other and the total magnetization would be zero. Thus, the short-range correlations lead to formation of magnetic patches, but the long-range order is strictly speaking not unravelled. In our case, we use the XMCD technique which measures an

ensemble of multiple spins (size of the X-ray beam is about 0.5 mm^2) and thus it averages over large areas that could, in principle, contain multiple domains. The nonzero remanence observed on such large area is definitely caused by long-range order.

3. Size of the exchange

The authors very nicely deduced the size of the exchange interaction from the XMCD loops. It would be nice to plot the expected behaviour as a line into the XMCD data of figure 2d to illustrate the degree of agreement. Further, it seems imperative to compare the experimentally deduced exchange constants to that of the ab-initio calculations. Actually, no quantitative data is given in the text on the latter.

Authors: We thank the reviewer for his suggestion. In the newly revised version we have implemented these fits to a Brillouin function in the mean field approximation. The obtained curves match the experimental data very well. We attribute the small discrepancy between fit and experimental data at small fields to an anisotropy term that is not included in our fitting function.

The reviewer writes that experimentally deduced exchange constants should be compared to those of ab-initio calculations. This was written already in the previous report of the reviewer, and we have already answered to this point in the previous reply: *“As for the DFT calculations, we do treat the full experimental system with the 2x2 arrangement of the molecules, and find self-consistently the ferromagnetic coupling. However, we cannot compute currently the individual exchange coupling of a molecule to next nearest neighbours, as for this we would need larger supercells. The calculation of the nearest neighbour exchange interaction is also nontrivial, as we have to enforce a ferromagnetic order to obtain the size of the interaction.”*

The reviewer writes that “Actually, no quantitative data is given in the text on the latter” (the ab-initio calculations). Respectfully, we would like to point out that there is a section “Density Functional Theory +U calculations” on pages 6 and 7, where many quantitative results of the calculations are presented. Also in the Methods section there is a comprehensive description of the calculations with six references. A further detailed part on the ab-initio calculations is given in the SI.

4. Strange XMCD data

I still do not understand the discrepancy in the presented data regarding the sub-lattice magnetizations. The XMCD spectra at nominally zero field in Figure 2b and d speak for an antiferromagnetic interaction at $B=0$. The data plotted in Fig. 2d shows ferromagnetic alignment at B near zero. This goes together with an uncertainty of 0.1T in the field calibration. Can the authors exclude a large susceptibility and the observed remanence is just due to the bad field precision?

Authors: The data presented in Figure 2d show an antiparallel alignment between moments on FeFpc and MnPc sub-lattices for fields smaller than 2 T, as is also clearly seen in Figures 2b and 2c. This suggests that if the large susceptibility of FeFpc would be responsible for the “jump” at zero field the FeFpc molecule would then have to behave diamagnetic (i.e. repelling magnetic field) for small fields ($B < 2 \text{ T}$) and paramagnetic (i.e. aligning along the field) for larger fields ($B > 2 \text{ T}$). We thus

have strong evidence that if the remanence would be caused by a bad field precision we would not be able to observe antiferromagnetic coupling for small non-zero fields. We have now provided further evidence of the antiferromagnetic interaction by presenting XAS/XMCD data taken at 1T and -1T for fields ramped down from 6.8T and -6.8T, respectively. These XAS/XMCD data taken at 2.5K at $\pm 1T$ have been added to Figure S2 in the revised SI.

5. Kondo temperature

I like the temperature dependent STS measurements to deduce the Kondo temperature by extrapolation. However, the STS data was obtained using a finite modulation which contributes to the spectral width of the features. It seems that this has not been taken into account. It would be great to do this in a proper way.

Authors: We thank the reviewer for this suggestion. In the revised manuscript and in the revised SI we have added newly obtained values of the Kondo temperatures that do include the finite smearing effects from the finite modulation voltage.

6. Ordering temperature

At 5 K, the remanence vanishes (Fig. S2), which is a nice confirmation of the claims. However, the manuscript does not specify, whether the XAS data were also taken after saturating the samples at high fields, as for the 2.5K data.

Authors: The data measured at $B=0T$ and at 5K were acquired after the magnetic field was ramped down (up) from 6.8T (-6.8T). We have provided the requested information in the revised SI, where we have added, in the caption of Fig. S2, that the XAS and XMCD data are taken at 5K in a magnetic field of 6.8 T.

We once again thank the reviewer for the helpful comments, which we believe to have fully addressed.

Reviewers' comments:

Reviewer #1 (Remarks to the Author):

I agree with referee 3 that there are still unclear issues regarding the actual magnetic ordering at zero field which were already pointed out by all the referees. This is a fact that I have been willing to overlook and (I am still) since the authors have made all possible efforts in this regard and the results are compelling in many other ways. Nevertheless, since their arguments on the experimental limitations are not totally convincing, I am going to take sides with referee 3 on his/her criticism about the calculations. Unless the authors can justify that such a calculation cannot be made, I do not see why they do not compute the ferromagnetic state which, it seems to me, requires a minimum amount of work since they already have the ferrimagnetic electron density. Reversing the spins to search for another magnetic solution is usually a trivial task. This would support a claim which is not fully supported by the experiment.

Reviewer #2 (Remarks to the Author):

To settle the dispute with Reviewer #3:

- Fig 2d /AFM ordering: Reviewer 3 is not convinced by the authors with respect to the experimental evidence to the AFM ordering of the molecular sublattices. Maybe the problem lies with the fact that it is difficult for the reader to map results shown in Figs 2b,c and in panels a,b, e,f of Fig. S2 into fig 2d.

If I take the data of Fig 2b,c then I see that when ramping the B-field from positive values to zero there is remanent magnetism on the MnPc lattice and AFM alignment on the FePc lattice.

In Fig. 2d however one might get a bit confused by the fact that there are also MnPc data points at $B=0+\epsilon$ that are ferromagnetically aligned to the FePc (and if the authors are right should come from ramping from negative B fields).

At ± 1 T the AFM alignment is also obvious as from S2 e,f. So up to this point the AFM coupling seems to be substantiated by the experiments.

However, I now don't understand where in Fig. 2d the datapoints of panels S2 a, b are plotted. In the caption of Fig S2 it is stated that "the magnetic moments of FePc (a) and MnPc (b) have changed their relative XMCD sign as compared to that observed at $B = 6.8$ T." The FePc signal is very hard to discern but it is clear that the MnPc moment has at -0.1 T followed the external field. Then I start to wonder where the MnPc data points in Fig 2d in negative magnetic field but with positive XMCD signal (i.e. with the same sign as at $B>0$ saturation) come from and hence are apparently coupled FM to the FePc lattice.

Additionally the authors should be more precise in giving an idea why the XMCD signal for $B>0$ of the MnPc goes down (also compared to the fit curve in 2d). The keyword "anisotropy" may not suffice.

If I compare the panels 2c and S2b I now also wonder why the XMCD signal is so clear and strong at $B=0$ and so weak at -0.1 T.

- DFT calculations: The authors should add a statement on pg.7 after the sentence describing the outcome of the DFT calculations that the entirely ferromagnetic state has a higher energy by xx meV. Alternatively, it should be clearly stated that the calculation relaxes into the AFM state even if started with FM configuration.

(Of course only, if that is the case. If this cannot be said, then the AFM ground state is uncertain and the calculations can only be used to show the magnetic moments and the suggested RKKY interaction between molecules as indicated by the induced spin polarization.

Reviewer #3 (Remarks to the Author):

Again, the paper improved. Some of my points have been fully answered, but I am sorry that I am still not satisfied. Two issues central to the claims remain.

1. Fig. 2d now shows the result of the fitting as solid lines. If indeed, the system would be a ferrimagnet, for $B \rightarrow 0$ the Fe and Mn moments should be antiparallel. This is also nicely shown by the solid curves from model calculations. Why does the experimental data deviate from this expected behaviour? The behaviour at low fields is actually the key to call something a ferrimagnet! Obviously, below about $\pm 0.5T$, the Mn moments start to vanish and even change sign near $B=0$. If I take the experimental data seriously, it would actually speak for a system in which the Mn spins show no remanence or possibly a ferromagnetic remanence (same sign as Fe moments). The authors write, that the deviation could be to anisotropy. I do not follow that statement. I cannot see that the anisotropy of the system would lead to a sign reversal of the magnetic moment of Mn in vanishing fields. I fully agree with the authors that for fields above 0.5 T, their model works fine, but to speak of a ferrimagnet with remanence is to make a statement about the behaviour at vanishing fields.

Generally speaking, the above problem should be obvious even to the general reader and will raise questions about the reliability of the presented work. The authors face this criticism during the review process but if ignored, they will face later with the whole community.

2. Still the authors claim that DFT predicts a ferrimagnetic situation of the exchange. What the authors actually show is a converged result of a DFT calculation showing magnetic moments in a ferrimagnetic arrangement (which is actually at odds with the data shown in fig. 2d for vanishing fields). They seem not to be willing to calculate the energy of the equivalent ferromagnetically ordered state. I agree that DFT can be difficult, but to just show a state after convergence strictly speaking does not provide the claims they make. The ferromagnetic state could be degenerate in energy or the DFT convergence might be trapped in a local minimum and other states might even be lower in energy. After all, the involved energies are really small and a careful study should check, whether the predictions given are actually stable, i.e. alternative states indeed have a higher energy. If the unit cell is the problem, they could still just use the force theorem to test the stability of their claims.

Similar to my statement above, this rather obvious problem will be noted by the general reader, as it has been a standard procedure in the last decade to compare energies of alternative states in order to make claims about exchange constants.

Reviewer #1 (Remarks to the Author):

I agree with referee 3 that there are still unclear issues regarding the actual magnetic ordering at zero field which were already pointed out by all the referees. This is a fact that I have been willing to overlook and (I am still) since the authors have made all possible efforts in this regard and the results are compelling in many other ways. Nevertheless, since their arguments on the experimental limitations are not totally convincing, I am going to take sides with referee 3 on his/her criticism about the calculations. Unless the authors can justify that such a calculation cannot be made, I do not see why they do not compute the ferromagnetic state which, it seems to me, requires a minimum amount of work since they already have the ferrimagnetic electron density. Reversing the spins to search for another magnetic solution is usually a trivial task. This would support a claim which is not fully supported by the experiment.

Authors: First, we would like to thank the reviewer for his/her strong support of our publication. As we mentioned before, the calculations concern a very large system, which had, notably, not been done before. However, in order to settle the issue we have performed numerical calculations for the ferromagnetic state, i.e. where the moments on the Fe molecules are reversed and parallel to the Mn moments. This state is calculated to have a selfconsistent total energy that is *higher* by 9K or $\Delta E = 0.78$ meV (per structural unit cell). This value is in very good agreement with the experiment. We have added these results to the main text and, with more detail, to the Supplementary Information.

Reviewer #2 (Remarks to the Author):

To settle the dispute with Reviewer #3:

- Fig 2d /AFM ordering: Reviewer 3 is not convinced by the authors with respect to the experimental evidence to the AFM ordering of the molecular sublattices. Maybe the problem lies with the fact that it is difficult for the reader to map results shown in Figs 2b,c and in panels a,b, e,f of Fig. S2 into fig 2d.

If I take the data of Fig 2b,c then I see that when ramping the B-field from positive values to zero there is remanent magnetism on the MnPc lattice and AFM alignment on the FePc lattice. In Fig. 2d however one might get a bit confused by the fact that there are also MnPc data points at $B=0+\epsilon$ that are ferromagnetically aligned to the FePc (and if the authors are right should come from ramping from negative B fields). At ± 1 T the AFM alignment is also obvious as from S2 e,f. So up to this point the AFM coupling seems to be substantiated by the experiments. However, I now don't understand where in Fig. 2d the datapoints of panels S2 a, b are plotted. In the caption of Fig S2 it is stated that "the magnetic moments of FeFPc (a) and MnPc (b) have changed their relative XMCD sign as compared to that observed at $B = 6.8$ T. " The FePc signal is very hard to discern but it is clear that the MnPc moment has at -0.1 T followed the external field. Then I start to wonder where the MnPc data points in Fig 2d in negative magnetic field but with positive XMCD signal (i.e. with the

same sign as at $B > 0$ saturation) come from and hence are apparently coupled FM to the FePc lattice.

Authors: We thank the reviewer for his/her suggestion to improve the quality of our manuscript. To address the raised issue we have improved Supplementary Figure 2 by adding XMCD peak height vs B curves with arrows showing the steps during the data acquisition.

The reviewer's question concerns the data shown in Figs. 2b,c and in Fig. 2d. To explain this, it is important to realise that two different measurement techniques are used for the data in these panels. Figs. 2b,c show XAS/XMCD measurements in *static* magnetic fields (0 T and 6.8 T). These measurements clearly show long-range order with antiparallel magnetic alignment of the molecules' spins at $B = 0$ T. The measurements shown in Fig. 2d have however been obtained with a different protocol. Here the magnetic field is *continuously* ramped while the XAS/XMCD measurement is performed at two different energies. Due to the continuous field ramping these are measured at slightly different magnetic fields (unlike the data in Figs. 2b,c). Since binning is performed to have the B -field in certain bins this leads to a difference of about 0.06 T for the measurement at the reference energy and at the background energy. In particular around zero field the measurement is susceptible to noise and the uncertainty in the actual magnetic field value. There are three data points for MnPc at around $B = 0$ T in Fig. 2d where this difficulty applies, and precisely these points are in the focus of this discussion. As asked above by the reviewer, we have explained where these MnPc data points come from. Also, as the measurement technique was different, one cannot map the data in Figs. 2b,c in a one-to-one fashion to those in Fig. 2d. However, we emphasize that in order to exclude confusion caused by these points we show explicitly in Figs. 2b,c that the XMCD signals of the MnPc and FePc molecules taken at static $B = 0$ T are *antiparallel*, demonstrating a ferrimagnetic ground state, which is further confirmed by the sum rule analysis (Supplementary Table 1) and the DFT+ U calculations.

To make the different experimental procedures clearer to the reader we have added a description of the measurement protocol to the Methods section and added a sentence to the main text about difficulties of the measurement at small fields within our experimentally reliable magnetic field resolution $\Delta B \sim 0.1$ T. A more detailed description has also been added to the Supplementary Information.

Additionally the authors should be more precise in giving an idea why the XMCD signal for $B \rightarrow 0$ of the MnPc goes down (also compared to the fit curve in 2d). The keyword "anisotropy" may not suffice.

Authors: We agree with the reviewer that anisotropy alone cannot explain the discrepancy. We therefore listed other possible explanations, e.g. the choice of the energy reference at which the XMCD peak height vs B curves have been recorded, which are however purely speculative as we are not aware of any publication, which could lead us towards resolving the issue.

If I compare the panels 2c and S2b I now also wonder why the XMCD signal is so clear and strong at $B=0$ and so weak at -0.1 T.

Authors: The XAS/XCMD spectra taken at $B = 0$ T (see Figure 2c) are composed of 16 individual spectra (8 taken after the magnetic field had been ramped down from 6.8 T to 0 T and another eight after the magnetic field had been ramped up from -6.8 T to 0 T). The spectra measured at -0.1 T (cf. Supplementary Figure 2b) are averaged from 4 spectra only.

- DFT calculations: The authors should add a statement on pg.7 after the sentence describing the outcome of the DFT calculations that the entirely ferromagnetic state has a higher energy by xx meV. Alternatively, it should be clearly stated that the calculation relaxes into the AFM state even if started with FM configuration.

(Of course only, if that is the case. If this cannot be said, then the AFM ground state is uncertain and the calculations can only be used to show the magnetic moments and the suggested RKKY interaction between molecules as indicated by the induced spin polarization.

Authors: As mentioned in the reply to reviewer #1 we have now carried out these calculations and find that the ferromagnetic state has a *higher* energy by 0.78 meV (per structural unit cell). This provides an independent support of the experimental observations. The text has been changed accordingly.

Reviewer #3 (Remarks to the Author):

Again, the paper improved. Some of my points have been fully answered, but I am sorry that I am still not satisfied. Two issues central to the claims remain.

1. Fig. 2d now shows the result of the fitting as solid lines. If indeed, the system would be a ferrimagnet, for $B \rightarrow 0$ the Fe and Mn moments should be antiparallel. This is also nicely shown by the solid curves from model calculations. Why does the experimental data deviate from this expected behaviour? The behaviour at low fields is actually the key to call something a ferrimagnet! Obviously, below about ± 0.5 T, the Mn moments start to vanish and even change sign near $B=0$. If I take the experimental data seriously, it would actually speak for a system in which the Mn spins show no remanence or possibly a ferromagnetic remanence (same sign as Fe moments). The authors write, that the deviation could be to anisotropy. I do not follow that statement. I cannot see that the anisotropy of the system would lead to a sign reversal of the magnetic moment of Mn in vanishing fields. I fully agree with the authors that for fields above 0.5 T, their model works fine, but to speak of a ferrimagnet with remanence is to make a statement about the behaviour at vanishing fields. Generally speaking, the above problem should be obvious even to the general reader and will raise questions about the reliability of the presented work. The authors face this criticism during the review process but if ignored, they will face later with the whole community.

Authors: We would like to bring to the reviewer's attention our comments to reviewer #2 where we tried to clarify the different measurement procedures that are at the origin of the whole issue.

2. Still the authors claim that DFT predicts a ferrimagnetic situation of the exchange. What the authors actually show is a converged result of a DFT calculation showing magnetic moments in a ferrimagnetic arrangement (which is actually at odds with the data shown in fig. 2d for vanishing

fields). They seem not to be willing to calculate the energy of the equivalent ferromagnetically ordered state. I agree that DFT can be difficult, but to just show a state after convergence strictly speaking does not provide the claims they make. The ferromagnetic state could be degenerate in energy or the DFT convergence might be trapped in a local minimum and other states might even be lower in energy. After all, the involved energies are really small and a careful study should check, whether the predictions given are actually stable, i.e. alternative states indeed have a higher energy. If the unit cell is the problem, they could still just use the force theorem to test the stability of their claims.

Similar to my statement above, this rather obvious problem will be noted by the general reader, as it has been a standard procedure in the last decade to compare energies of alternative states in order to make claims about exchange constants.

Authors: We do agree with the reviewer that a comparison of energies has been done in the last decade, but, notably, for much smaller systems. We have already explained in the previous replies that the system under consideration is very large, and similar calculations had never been done before. Nonetheless, as mentioned above, we have now performed these calculations and find that the ferromagnetic state is higher in energy by 0.78 meV or 9 K (per structural unit cell), in good agreement with the experiment.

The new calculations are now mentioned in the main text and in the SI. In addition, we have added two sentences to the theory part in which we mention the theoretical prediction of the appearance of a vortex phase with zero mean magnetisation, the occurrence of which is however not supported by our XMCD data that show nonzero mean sublattice magnetisation.

All changes made to the main text and Supplementary Information are highlighted with yellow colour.

Having addressed the two points of the reviewers, we hope that our manuscript can now be recommended for publication.

REVIEWERS' COMMENTS:

Reviewer #2 (Remarks to the Author):

The authors have amended the paper and substantiated the original claims of the manuscript. It is very unfortunate that the authors have only now added the crucial experimental protocols. From my point of view the paper contains a lot of exciting experimental findings that deserve publication in Nat.Comm..